# Accessibility measurement of highway transportation networks based on closeness-accessibility

Yuanyuan Zhang[1,2¤a], Weidong Song ᴵᴰ[1*], Jinguang Sun[3¤b], Peng Dai[4]

1 School of Geomatics, Liaoning Technical University, Fuxin, China, 2 School of Civil Engineering, Liaoning Technical University, Fuxin, China, 3 School of Electronics and Information Engineering, Liaoning Technical University, Huludao, China, 4 Department of Basic Teaching, Liaoning Technical University, Huludao, China

¤a Current Address: School of Geomatics, Liaoning Technical University, Fuxin, China
¤b Current Address: School of Electronics and Information Engineering, Liaoning Technical University, Huludao, China
* songweidong@lntu.edu.cn

## Abstract

Accessibility is a central concept in transport geography research. Quantitative assessment of highway network accessibility aids in assessing road network efficiency, urban layout optimization, functional zoning and sustainable development. Current metrics for assessing accessibility primarily concentrate on examining the relationship between land use and transportation networks, overlooking the influence of trip demand and node centrality on highway network accessibility. To clarify these impacts, borrowing from concepts in network science, we propose a new centrality measure called closeness-accessibility. This approach utilizes closeness centrality of network nodes as a weight to assess the interaction potential between nodes, enhances the gravity-based accessibility of demand points based on trip time, and incorporates network connectivity to evaluate the impact of travel demand on highway network accessibility and the significance of node centrality non-uniformity. Liaoning Province provides an empirical case study to showcase the usefulness of closeness-accessibility. The findings indicate a positive correlation between the gravity-based and closeness-accessibility generated by various β values, showing a linear relationship between the two. Closeness-accessibility takes into account the number and centrality of demand points in influencing travel demand, enhances the calculation of the shortest paths of the spatial location of the demand nodes, and improves the accuracy of accessibility calculations. Nodes with higher centrality exhibit greater accessibility, leading to clearer delineation of accessibility levels for demand points and highway network across different regions. In Liaoning Province, the areas with high demand attraction and highway network accessibility are concentrated in the coastal area and the provincial capital center, while accessibility services in western part are inadequate. Although all grades of highway accessibility service

**Data availability statement:** All data used in this article have all been published on the figshare repository, and the URL: https://doi.org/10.6084/m9.figshare.29973184.

**Funding:** National Natural Science Foundation of China 42071343. The funders had no role in study design, data collection and analysis, decision to publish, or preparation of the manuscript.

**Competing interests:** The authors have declared that no competing interests exist.

are high, higher-grade highways sometimes show lower accessibility. This research offers a methodological tool for transportation experts and holds significant importance in the scientific and effective evaluation of transportation system performance.

## Introduction

The development of a complex highway transportation network in China has led to increased urban connectivity. National highways, Provincial highways, and County highways exhibit distinct transportation characteristics, serving as vital links between cities. An efficient highway network reduces trip distances, ensures passenger and cargo safety, meets trip demands, and significantly impacts the economic development of cities. Transportation network accessibility measures the ease of trip from origin to destination within a specific transportation system, serving as a key indicator for assessing network design, regional development potential, and overall transportation system performance.

Accessibility is a crucial evaluation metric for highway network planning schemes. Scholars globally have extensively examined highway network accessibility. Gutiérrez (1996) [1] focused on the European highway network to investigate its influence on traffic. Subsequent studies on accessibility have garnered interest from scholars in urban planning, transportation geography, economic geography, health geography, regional structure and space research [1–5].

The accessibility of regional transportation and public service facilities significantly impacts individuals' trip convenience and well-being. Recently, scholars have shown great interest in calculating and evaluating the accessibility of social public services, including healthcare, education, and cultural tourism, etc [6]. Many studies have utilized GIS spatial analysis to assess the spatial accessibility of facilities across various regions in China. Nevertheless, it is essential to note that accessibility analysis should not only consider spatial factors but also incorporate temporal considerations.

Spatial-temporal accessibility is a comprehensive indicator for evaluating the service level of transportation systems. It is based on the analysis of individual spatial-temporal behaviors and prioritizes meeting travelers' demand for reaching activity destinations. The advancements in transportation infrastructure and services play a crucial role in enhancing overall accessibility levels.

Quantitative assessment of highway network accessibility is necessary. Current metrics for assessing accessibility primarily concentrate on examining the relationship between land use and transportation networks, overlooking the influence of trip demand and node centrality on highway network accessibility. Neglecting these factors may hinder the highway network's capacity to support future resilient infrastructure development, leading to operational instability and inefficiency.

The study aims to quantify the impact of different travel demand patterns on the overall accessibility of the highway network. By comparing network performance in a real network, the importance of node centrality non-uniformity is evaluated. The proposed closeness-accessibility modeling bridge spatial interaction theory and

network science. Our case study focuses on the three-level highway network of national, provincial and county highways in Liaoning province, utilizing only structural data and excluding any information about traffic flow.

The rest of this paper is organized as follows. In the following section, we briefly review the complex network model and accessibility analysis methods. Next, we explain accessibility framework, expound construction of neighbor topology network, elaborate the research methodology that helps illustrate how closeness-accessibility is calculated and what it represents. This section is followed by a case study of Liaoning province where the corresponding closeness-accessibility results are utilized for accessibility analysis purposes, as a potential use case. We conclude the paper by discussing the potential and limitations of our proposed measures and some directions for future research.

## Literature review

### Highway complex network model

Transportation networks can be classified into four main categories: natural networks, mesh networks, tree networks, and circular radial networks [7,8]. Early studies on transportation networks applied complex network theory to analyze their topological structures [9–12].

The primary research methodologies of complex networks theory are based on graph theory. Construction approaches predominantly encompass primal approach [13] and dual approach [14]. The primal approach represents networks by treating links as edges and nodes as vertices, thereby translating real-world geographical data on road intersections and segments into a road transportation network. Conversely, the dual approach abstracts roads within a transportation network as nodes in a topological framework, with road intersections being depicted as edges signifying topological connections between roads [15–16]. While effective in illustrating link interactions, this method may be limited in analyzing the spatial structure of extensive networks and may overlook road attributes. An analysis was conducted on the structural characteristics of the road network under different network construction modes, including the degree and degree distribution of network nodes, connectivity, centrality, average path length and diameter, clustering coefficient, and power-law distribution, etc. [17].

Garrison and Marble (1962) [18] utilized graph theory to establish the alpha (α), beta (β), and gamma (γ) connectivity indices for highway networks. Kansky (1963) [19] expanded on the connectivity index to introduce novel indicators across the transportation network, investigating factors influencing economic development. The application of graph theory in transportation research is widespread. Musso and Vuchic (1988) [20] devised transport-specific indicators, while Gattuso and Miriello (2005) [21] categorized 13 distinct transportation networks using existing indicators and introduced "node ranges of interest." Spatial accessibility is evaluated based on distance attenuation effects and the populations accessing a specific point of interest (POI) [22–25]. Radke and Mu (2000) [26] introduced a gravity-based measure to equitably distribute social programs. The interconnectedness of interaction potential and network support in geographical research is widely acknowledged, with network analysis playing a crucial role. Examples of network analysis applications include vulnerability assessment [27,28], research on spatial inequality [29], and urban structure classification [30].

### Accessibility analysis methods

Accessibility has been a recurrent topic in the literature since Hansen (1959) [31] seminal work which defined it as "the potential of opportunities for interaction". Other definitions of the concept define it as "the ease with which any land-use activity can be reached from a location using a particular transport system" [32] or "the benefits provided by a transportation/land-use systems" [33]. Undoubtedly, urban planners, geographers, and transportation professionals have used this concept widely in the last decades to analyze the Land-Use Transport Interaction cycle [34]. Geurs and Van Wee (2004) [35] identified four distinct categories of measuring accessibility: infrastructure-based, location-based, person-based, and utility-based.

Infrastructure-based measurements is widely used in transport planning. This category is related to the level of service of transportation infrastructure, e.g., travel speeds on the network [36,37]. Location-based measures analyzes the interaction between land uses and the transportation network. This category can be divided into two subgroups. The first focuses on distance-based measures (isochrones analyses), in which the connectivity between two zones is given by distance measures between them. These measures can be the Euclidean distance, Manhattan distance, trip time, or transportation cost access opportunities [35,38,39]. The second sub-group is the potential accessibility models, or gravity-based models, which compute the number of potential opportunities by origin to all destinations. This category of accessibility is widely used in urban planning and geographical studies.

Person-based measures analyzes accessibility at an individual level-"the activities an individual can participate at a given time" [35]. This category is based on the seminal work of Hägerstrand (1970) [40] who measured the travel limitations of individuals due to distance and travel time constraints. Other authors that have worked in this category are Burns (1980) [41], Miller (1999) [42], and Reckera et al. (2001) [43]. Utility-based measures has its foundations in economic theory. It understands accessibility as a byproduct of a set of transportation choices. Two types of measurements have been used for this category: doubly constrained entropy models [44] and logsum indicators [33].

The gravity-based accessibility [45–47] reflects the degree of matching between potential demand and facility supply. Thomas et al. (2003) [48] analyzed freight accessibility in rail, road, and waterways in Belgium. The authors used gravity-based measures weighing the nodes in the network by population and economic activity. The study shows there is a positive correlation between population and transportation infrastructure; it also shows that economic activities are less related to transportation infrastructure.

Salas-Olmedo et al. (2015) [49] improved the gravity-based model to examine the effects of new road transportation infrastructure on the accessibility of the European Union. Rodolfo et al. (2020) [50] proposed a gravity-based model utilizing the trip-impedance method with friction surface datasets and kernel density maps. Giannotti et al. (2022) [51] analyzed the potential biases in utilizing mobility-based gravity measurements to identify potential inequalities in accessibility.

Centrality analysis is a key issue in network studies and serves as a method to assess the significance of nodes within a network. The commonly used centrality measure is the degree centrality of a node, which quantifies the number of edges connected to a node [52]. Betweenness centrality evaluates the number of shortest paths that pass through a node [53]. Closeness centrality measures how close a node is to other nodes [54]. In network analysis, centrality is related to the paths of node interactions, and various centrality perspectives can supplement the accessibility analysis [55–57].

Georgios et al. (2020) [55] proposed a new centrality measure called betweenness-accessibility, which was useful to estimate the impacts of accessibility on networks as potential for interaction is reflected on them. The new centrality formulation allows overcoming previously identified shortcomings of traditional betweenness measures, resulting in a measure tailored for networks with heterogeneous interaction levels. A new set of indicators combining the concepts of centrality and gravity-based accessibility was introduced and expanded the potential of both centrality analysis and accessibility analysis.

## Research methodology

### Accessibility framework

Fig 1 shows the methodological flowchart and it is divided into three sections; first: required databases, second: inputs from those databases, and third: processes to calculate closeness-accessibility. The first database is responsible for storing and managing pertinent data, including indicator data, POI data, national, provincial, and county highway network data. The second data input involves data processing and analysis within the database. It is used to calculate the trip time and distance for the highway network data. GIS interpolation analysis is conducted on the population, GDP and car ownership of the road facility points and demand points. The third calculation process involves using GIS's neighbor analysis

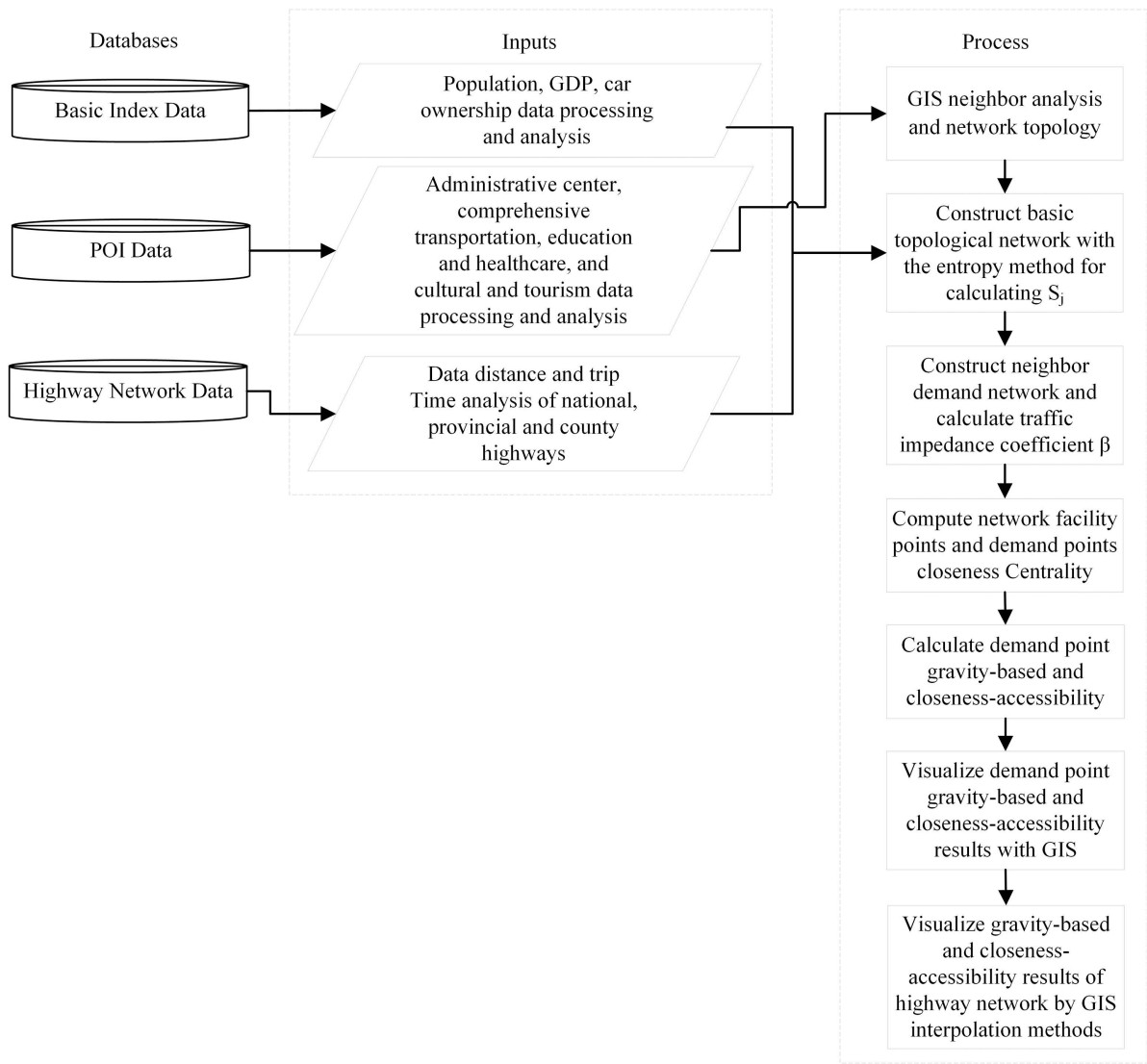

**Fig 1. Framework flowchart of the methodology.**

in conjunction with the NetworkX library in Python to construct the highway transportation network and calculate the closeness centrality of the network nodes. By using the entropy method to compute the weighted sum of population, GDP, and car ownership of highway facilities, the service capacity ($S_j$) of highway facilities can be derived. The median and mean trip times are then utilized to calculate the traffic impedance coefficient ($\beta$). Lastly, once both accessibility metrics are calculated, visualize the results using any GIS software.

## Construction of neighbor topology network

**Basic topology network.** The basic topology layer describes the relationships between different elements in the highway network. In this study, the primal approach is used to construct a undirected weighted network based on the attributes of road sections. Taking the endpoints and intersections of the road sections as the topological vertex set; the trip time as the

weight; national highways, provincial highways and county highways are constructed as the edge set. Therefore, the primal approach should be modified as follows:

Step 1: Construct the topological vertex set V with the endpoints and intersections of the road sections in the highway network as the nodes of the topological network.

Step 2: Construct the edge set E with the road sections in the highway network as the edges of the topological network.

Step 3: Construct an undirected weighted network with road trip time as the weight and obtain the weight set W.

**Neighbor demand network.** Identify key demand attraction points (administrative center, comprehensive transportation, education and healthcare, and cultural and tourism) to construct the demand vertex set. Connect the demand nodes to the basic topology network, as shown in Fig 2. The specific steps of constructing a neighbor demand network are shown as follows:

Step 1: Traffic zones are divided by the county administrative regions, and the centroid points of zones are determined to construct the demand vertex set V*.

Step 2: Identify Points of Interest (POI) such as comprehensive transportation, education and healthcare, and cultural and tourism, and include them in the demand vertex set V*.

Step 3: Connect the centroid points and POI to the nearest edge in the basic topology network, constructing the edge set E*.

Step 4: Calculate the trip time of neighbor edges to obtain the weight set W*.

Step 5: Construct an undirected weighted network G* = (V*, E*, W*).

## Gravity-based accessibility

The gravity-based accessibility is also referred to as the potential accessibility model. This paper is based on the gravity-based model proposed by Weibull [47]. This model quantifies the spatial or temporal accessibility of demand attraction points. Eq. 1 shows its mathematical formulation.

$$A_i^D = \sum_{j=1}^{n} \frac{S_j d_{ij}^{-\beta}}{V_j}$$

(1)

where $A_i^D$ is the gravitational accessibility index; n is the number of facilities; $d_{ij}$ is the distance (or time) between demand point $i$ and facility point $j$; and $S_j$ is the service capacity of facility point $j$. The larger the $A_i^D$ value, the better the accessibility. $V_j$ is gravitational potential energy. Eq. 2 shows its mathematical formulation.

$$V_j = \sum_{k=1}^{m} P_k d_{kj}^{-\beta}$$

(2)

where m is the number of demand points; $P_k$ is the population at the demand point; $d_{kj}$ is the distance (or time) between demand point $k$ and facility point $j$; $\beta$ is the traffic impedance coefficient.

The calibration of the $\beta$ parameter is a challenge when computing gravity models, The literature recommends an iterative process based on observed trip cost data to calibrate this parameter for gravity models [58]. However, such data is currently unavailable, making the calibration process unfeasible. The method used by Östh et al. (2014) [59] for calculating the $\beta$ value is cited, as shown in Eq. 3.

$$-\beta = \frac{\ln(0.5)}{M}$$

(3)

Where M represents the median distance or time traveled, $\beta \in [0,1]$ signifies the influence of traffic impedance on trip distribution. $\beta$ value of 0 indicates that travel cost does not affect travel distribution or accessibility. Conversely, $\beta$ value of 1 signifies that cost is the primary determinant in the accessibility metric. This parameter is closely linked to demand elasticity.

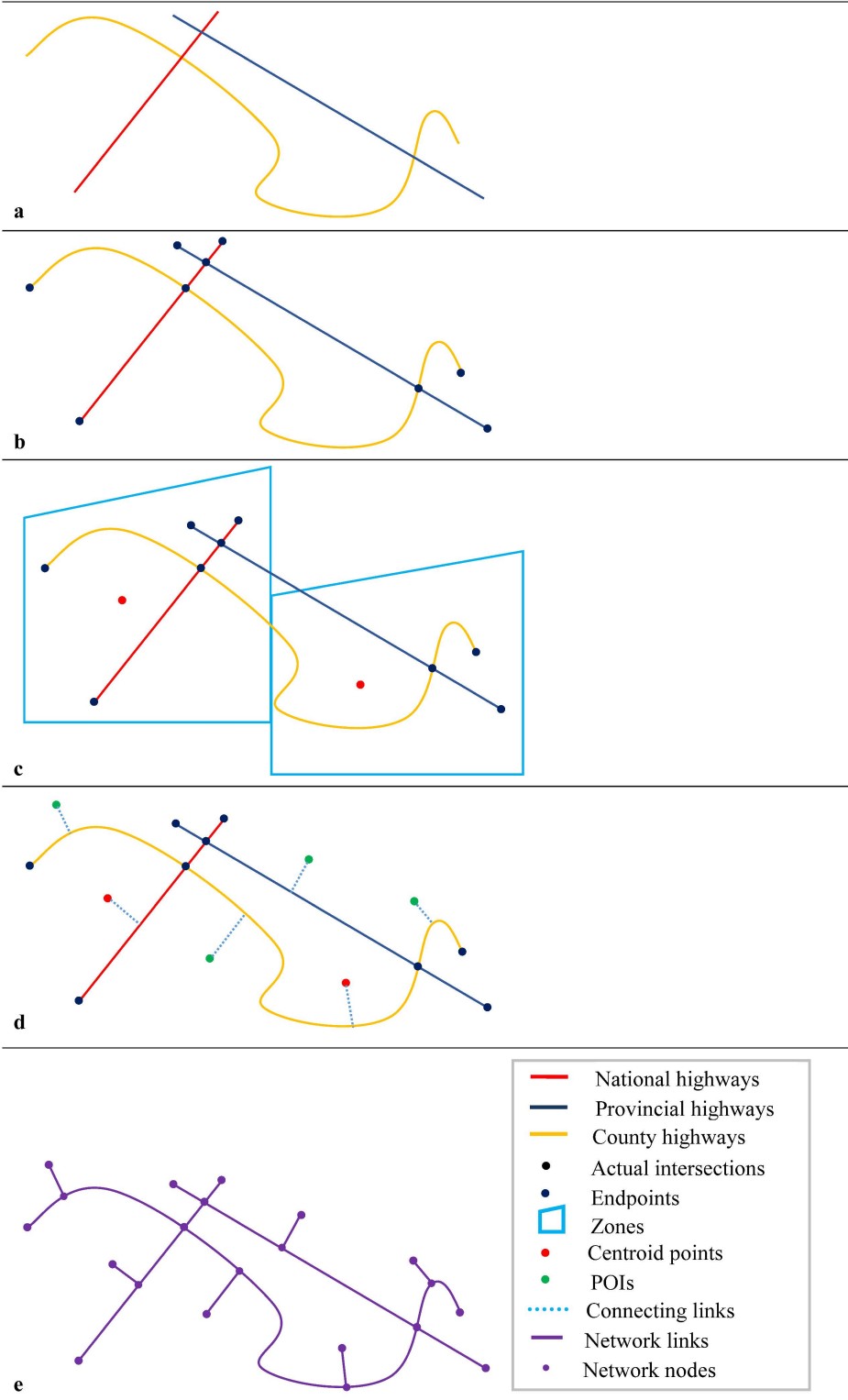

**Fig 2. Neighbor demand network construction diagram.**

## Closeness-accessibility

**Closeness centrality.** Centrality indicators are essential for analyzing transportation networks, providing insights into network topology [28,60,61]. Closeness centrality assesses the closeness of a node to others. It measures the reciprocal of the average shortest path distance between this node and all other nodes in the network. Eq. 4 presents the normalized calculation for centrality. Closeness centrality values range from 0 to 1. The higher the value, the smaller the sum of the distances from this node to all other nodes in the network.

$$Cc(v_x) = (N-1)/\left[\sum_{y=1,y\neq x}^{N-1} d_{xy}\right]$$

(4)

where $Cc(v_x)$ is closeness centrality of node in the graph, $N$ is the total number of nodes in the graph, $x$ is the target node to be computed, $y$ is any node connected to $x$ by an edge (excluding $x$), $N-1$ is the number of $y$, and $d_{xy}$ is the shortest path distance from $x$ to $y$.

For an unconnected graph, if there are no edges between nodes on different connected components, the distance between them is infinite, and the closeness centrality cannot be calculated. Therefore, in unconnected undirected graphs, the above definition needs to be modified. Wasserman and Faust (1994) [62] proposed an algorithm for calculating the closeness centrality of nodes, which expresses the calculation of the node closeness centrality within the connected components, and then weights the calculation results according to the number of nodes in the connected components, as shown in Eq. 5.

$$Cc_F(v_x) = \frac{n-1}{N-1}/\left[\frac{n-1}{\sum_{y=1,y\neq x}^{n-1} d_{xy}}\right]$$

(5)

where $Cc_F(v_x)$ is closeness centrality of node in the unconnected graph, $N$ is the total number of nodes in the graph and $n$ is the number of nodes in a connected component. $\frac{n-1}{\sum_{y=1,y\neq x}^{n-1} d_{xy}}$ expresses the closeness centrality of a node within a connected component, which is the reciprocal of the average shortest path distance between this node and other nodes in the connected component. $\frac{n-1}{N-1}$ is used to normalize the calculation results to the range of [0,1], so as to facilitate the comparison of the node closeness centrality of different connected components, and fully reflect the closeness of the nodes in the subgraph rather than the entire graph.

**Closeness-accessibility.** The interactions among nodes in the gravity-based accessibility are influenced by the attributes of the nodes, including factors like population and economic opportunities. Given the spatial heterogeneity of population distribution and opportunities, and the varying significance of nodes, the gravity-based model can be improved by embedding the closeness centrality weight to express the non-uniform influence of node centrality. Additionally, the spatial position of node centrality is used to modify the gravity-based model based on trip time calculation to achieve comprehensive spatial-temporal analysis of accessibility. Eq. 6 shows its mathematical formulation.

The assumptions and limitations of the model are that the calculation of trip time does not take into account for the influence of dynamic and real-time data such as congestion, road quality, temporal traffic variability, driving habits, weather, etc.

$$AD_i = \sum_{j=1}^{n} \frac{S_j t_{ij}^{-\beta} Cc(v_j)}{VD_j}$$

(6)

where $AD_i$ is the closeness accessibility index; $t_{ij}$ is the time between demand point $i$ and facility point $j$; $Cc(v_j)$ is closeness centrality of facility point $j$. If the network is connected, use Eq. 4 to calculate the method, otherwise use Eq. 5 to calculate, and involve the calculation of the universal shortest path distance between two nodes.

The service capacity ($S_j$) of facility j is determined as the weighted sum of population, GDP, and car ownership, as shown in Eq. 7. Various methods exist for assigning weights, including the analytic hierarchy process (AHP) [63], equal weighting, correlation analysis (CAM) [64], deviation maximization method (DMM) [65], and entropy method (EM) [66]. The entropy method (EM) is a diversity-based approach for determining weights by assessing the diversity of attribute data across different scenarios [67]. Compared to subjective weighting methods represented by the AHP, EM demonstrates greater objectivity, heightened sensitivity to extreme values, data-driven characteristics, and computational simplicity. Hence, in this study, EM is adopted to establish the model weights.

$$S_j = \omega_{jpop}\text{pop} + \omega_{jGDP}GDP + \omega_{jcar}car \tag{7}$$

where $VD_j$ is gravitational potential energy, Eq. 8 is the calculation method; $t_{kj}$ is the time between demand point $k$ and facility point $j$; $Cc(v_k)$ requires the closeness centrality of node k, if the network is connected, use Eq. 4 to calculate, otherwise use Eq. 5. The calculation of the distance between two nodes is based on the shortest path distance.

$$VD_j = \sum_{k=1}^{m} P_k t_{kj}^{-\beta} Cc(v_k) \tag{8}$$

The calculation method of the $\beta$ value is as shown in Eq. 3. This study calculates various $\beta$ values for the median and mean of trip time to obtain comparable accessibility analysis results under the influence of demand elasticity.

This study adopts the trip time to calculate the gravity-based accessibility and closeness-accessibility. Fig 3 illustrates the comparison process. Gravity-based accessibility assesses the spatial or temporal accessibility of demand attraction points and analyzes the potential interactions between nodes, which are influenced by the attributes of the nodes. It does not take into account the differences in the importance of node spaces and the resulting non-uniform effects.

The closeness-accessibility model incorporates node closeness centrality, fully demonstrating the non-uniform impact of node spatial importance on accessibility. It combines spatial distance and trip time to comprehensively calculate the accessibility of demand points. At the same time, it takes into account network connectivity and the characteristics of the network structure to accurately and reliably reflect the accessibility level of the research object.

## Study area and data

### Case study

The case study focuses on the three-level highway network of national, provincial and county highways in Liaoning province. The three-level highway network play a significant role in the regional economic development by linking the economic centers, transportation hubs, tourist destinations, educational and medical facilities, and historical and cultural sites etc. This network facilitates personnel mobility, material exchange and technological cooperation, and promotes the economic development of each city. Meanwhile, national, provincial and county highways serve as the crucial links between expressways and rural roads, connecting various cities with rural areas to offer convenient transportation services and meet people's travel needs.

Liaoning Province is the only province in Northeast China that has both coastal and border areas. It covers an area of 148,700 square kilometers and has 14 prefecture-level cities. As of 2020, the estimated population reached 41.659 million, with a GDP of CNY 251.15 billion. The total length of highways was 129,928 kilometers, including 9,957 kilometers of national highways, 10,251 kilometers of provincial highways, and 8,667 kilometers of county highways. The car ownership was 9.343 million. The data sourced from the Liaoning Province Statistical Yearbook for 2021 (https://tjj.ln.gov.cn) [68].

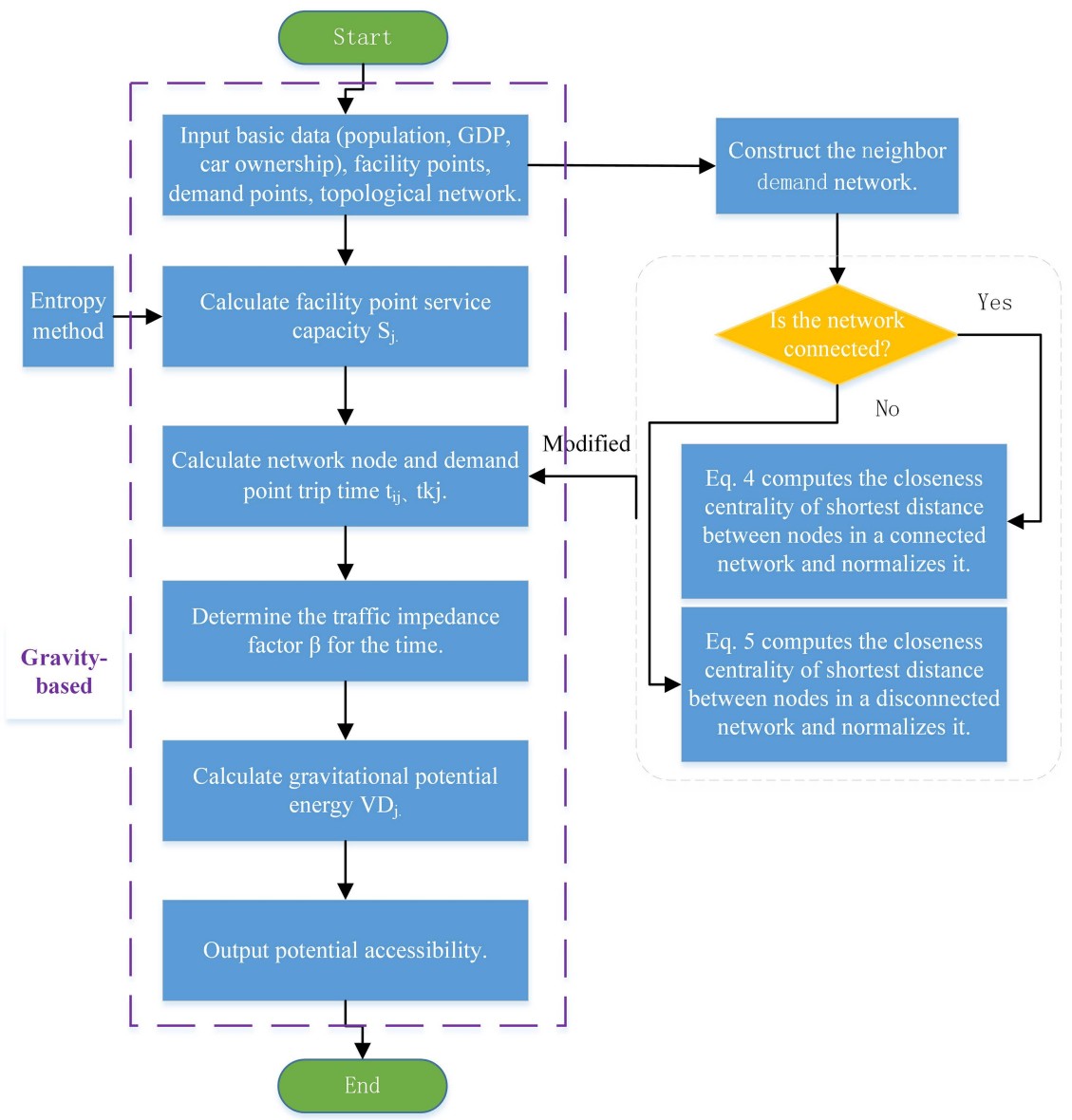

**Fig 3. Process comparison of gravity-based and closeness-accessibility models.**

## Data

This study examines five indicators of county-level data from the statistical yearbooks of Liaoning Province and its constituent cities in 2021, including population, GDP, highway mileage, car ownership, and land area. Population data is obtained from the data of street offices in the seventh national population census of China in 2020.

The analysis primarily utilizes county-level administrative data, national, provincial, and county highways, and regional POI data sourced from OpenStreetMap (OSM) in 2020. Using QGIS, maps depicting the distribution of population, GDP, car ownership, and highway network density were generated (Fig 4). Fig 4 shows the distribution of indicator data for each county and district in Liaoning Province. Regions with higher population density exhibit broader ranges, while those with elevated economic development levels, car ownership rates, and highway network density display narrower ranges.

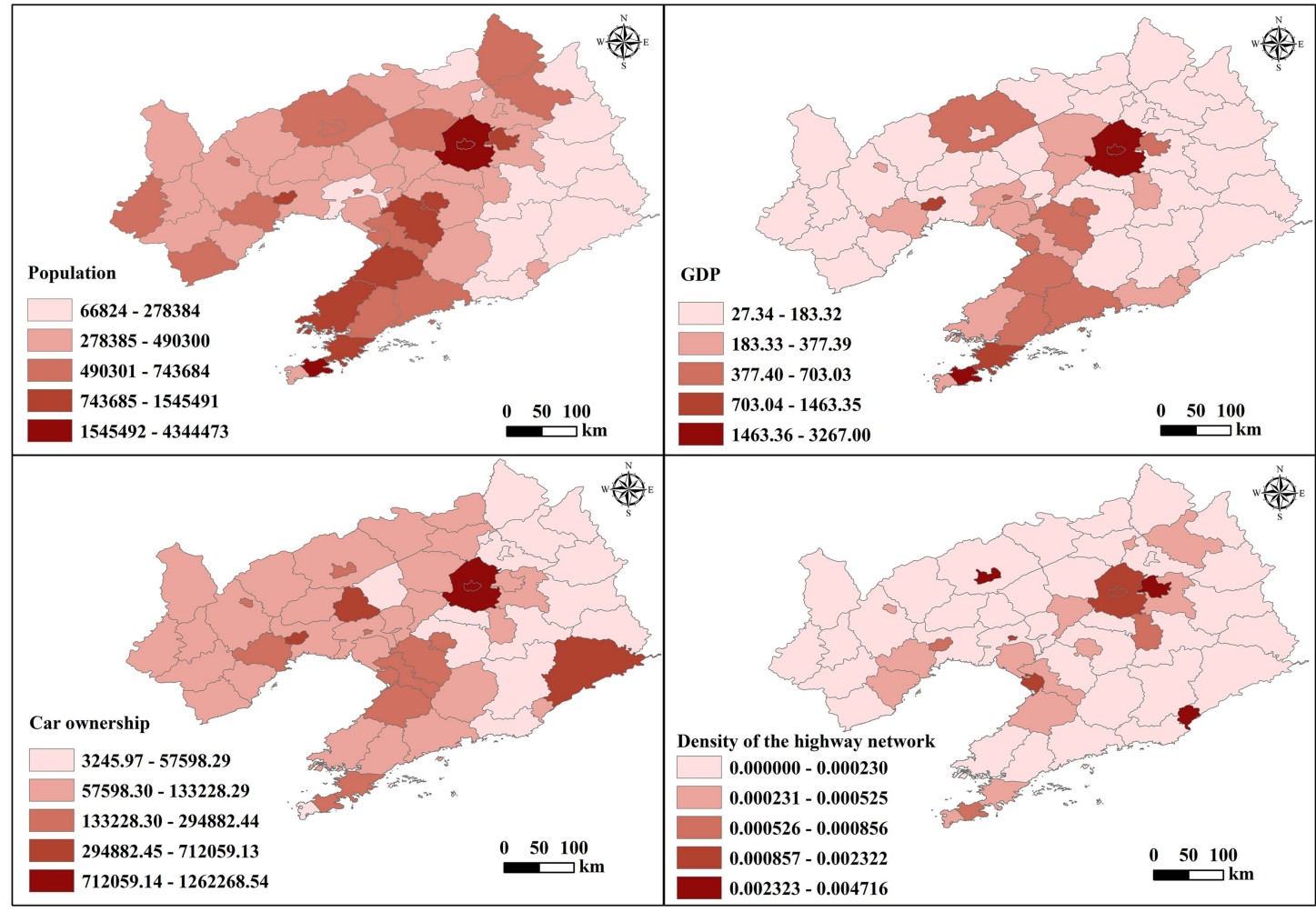

**Fig 4. Distribution of Indicator Data by County and District in Liaoning Province.** Base map data from GADM (https://gadm.org), version 4.0.

## Results

The near analysis in ArcGIS is used to obtain the nearest connection lines between centroid points and POIs. A transportation network dataset for Liaoning Province is constructed. Combined with the network analysis method and interpolation analysis in the GIS software, the trip time of the near road network is calculated.

Referencing the "Technical Standards for Highway Engineering of the People's Republic of China (JTGB01–2013)," combined with the research of relevant scholars and the current situation of Liaoning Province, the trip speeds is determined according to different highway classifications. The speed for national highways is 80 km/h, for provincial highways it is 60 km/h, and for county highways it is 40 km/h. In ArcGIS, the highway speeds are added to the highway attributes, and then the trip time each highway segment is calculated.

The Python library networkx 2.8.7 is utilized to construct the neighbor demand network for highway transportation, comprising 4605 nodes and 5460 edges where the edge weights represent trip time. The network is a connectivity graph, and the closeness centrality of its nodes is calculated. The Dijkstra algorithm is used to obtain the shortest path distances between all nodes in the network, and Eq. 4 is used to calculate the closeness centrality of the nodes in the highway neighbor demand network.

Fig 5(a) shows the distribution of $Cc(v_j)$ exhibiting an approximate normal distribution with a mean of 0.0258 and a maximum value of 0.0262, suggesting a significant distance between highway nodes. Fig 5(b) illustrates the distribution of $Cc(v_k)$, also displaying an approximate normal distribution with an average value of 0.0250 and a maximum of 0.0255, indicating considerable distance between demand nodes and low closeness between nodes. There are differences in the centrality values, demonstrating non-uniformity.

The median and mean of the network trip times are utilized to calculate traffic impedance coefficient β, resulting in values of 0.41 and 0.13, respectively. The service capacity of highway nodes is obtained by incorporating population, GDP, and car ownership using the entropy method. Fig 6 shows the edge histograms of the gravity-based and closeness-accessibility of the demand points. A positive correlation between gravity-based and closeness-accessibility is observed in the scatter plots. The histograms reveal the data distribution for both types of accessibility. The trip time impedance has a greater weight in the results when β = 0.41, with data distributions approximating normality. Variations in accessibility values were noted across different β values. Strong positive correlation is confirmed through linear fitting, underscoring the reliability and precision of the closeness-accessibility model's computations.

The accessibility of demand points can be treated as an ordinal scale and is useful to determine a ranking of accessibility among the zones in the study area. The higher the accessibility, the higher the zone's accessibility in comparison to the others. For this reason, Fig 7 shows the comparative analysis results of gravity-based and closeness-accessibility of the zones in the study area. The results reveal similarities in the accessibility of the gravity-based and closeness-accessibility of the different regions. For the two β values, the areas with higher accessibility are predominantly situated in the coastal area of southern Liaoning, characterized by high population density, advanced economic development, increased car ownership, and a dense highway network (Fig 4). Conversely, regions with lower accessibility levels are clustered in western Liaoning, marked by high population density, lower economic development, increased car ownership, and reduced highway network density (Fig 4).

The accessibility levels of demand points in different regions reflect the attractiveness of these areas. The coastal area of southern Liaoning and the provincial capital exhibit strong appeal, representing 19.84% and 16.89%, respectively. These regions offer a higher concentration of amenities in healthcare, education, culture, and tourism, along with well-developed transportation infrastructure, totaling 34.57%. This is in line with the overall development situation of Liaoning Province (https://tjj.ln.gov.cn). Conversely, western Liaoning displays lower appeal due to limited economic advancement and highway network density (Fig 4). The average proportion of overall regional attractiveness with a weak level is

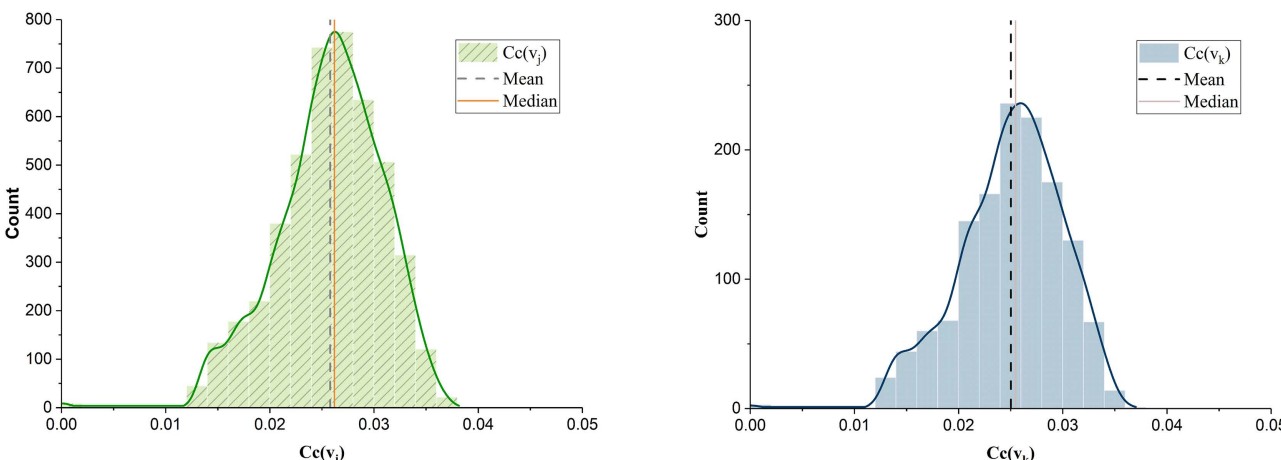

**Fig 5. Distribution of *Cc(vj)* and *Cc(vk)* of neighbor demand network in liaoning.**

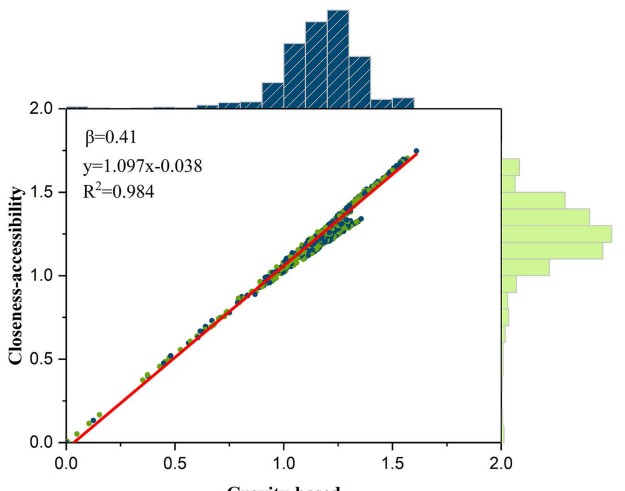
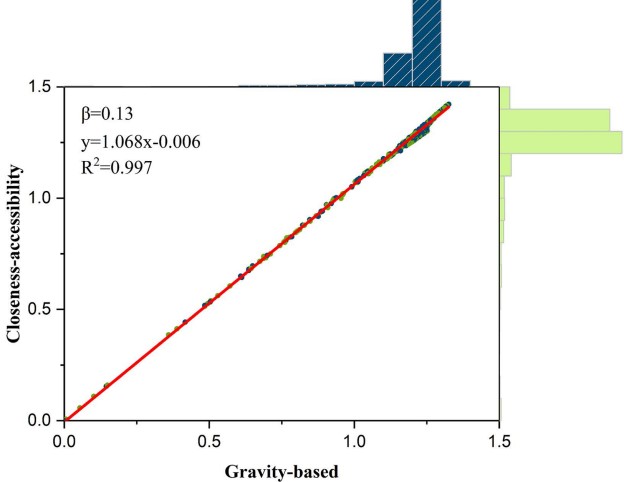

**Fig 6. Gravity-based and closeness-accessibility of demand points with histograms.**

42.53%. Utilizing the same symbol for identical β values, the accessibility assessment incorporating spatial centrality and temporal characteristics reveals that factoring in the number and centrality of demand points enhances the accuracy of the shortest path calculations between demand points. These findings from the accessibility analysis offer clear insights into the attractiveness levels of demand points in each region.

The accessibility level of the highway network between regions in the study area can be assessed using GIS inter-polation analysis, in conjunction with the accessibility of the demand points. The higher the accessibility, the higher the road accessibility of that area in comparison to the others, as shown in Fig 8 and Fig 9. Fig 8 displays an edge histogram illustrating the gravity-based and closeness-accessibility of the highway network. These two variables exhibit a linear fit, demonstrating a strong positive correlation between them.

Fig 9 shows the results of gravity-based and closeness-accessibility of the highway network in the study area. For the same β value, the results of the regional accessibility measurement of the highway network considering the combination of node spatial centrality and temporal characteristics show that the attractiveness generated by the differences in the number of demand attraction points, the centrality of road nodes, and the density of the road network is different. The results indicate the accessibility level of the highway network in each region.

The regions with high accessibility of demand points (Fig 7) have a relatively high level of highway network acces-sibility, indicating a correlation between trip demand points and highway network accessibility. The differences in node centrality result in a non-uniform spatial distribution of highway network accessibility. The regions around nodes with high centrality have higher accessibility, while regions around nodes with low centrality have lower accessibility [69]. Moreover, regions characterized by higher highway density (Fig 4) correspondingly exhibit enhanced accessibility.

Variations in β values reveal disparities in accessibility levels across regional highway networks. Closeness-accessibility analysis offers insights into the comprehensive influence of population, GDP, car ownership, highway network density, and trip demand on regional highway network accessibility. High accessibility is observed in coastal areas and provincial capital centers characterized by elevated levels of population density, economic development, car ownership, and highway network density (Fig 4). Conversely, low accessibility is prominent in western Liaoning, where these factors are comparatively low (Fig 4). Findings indicate that regions with dense populations, high GDP, elevated car ownership, and dense highway networks exhibit higher highway network accessibility levels [70–72]. Disparities in population distri-bution and accessibility levels are primarily concentrated in the western region, while differences in GDP development

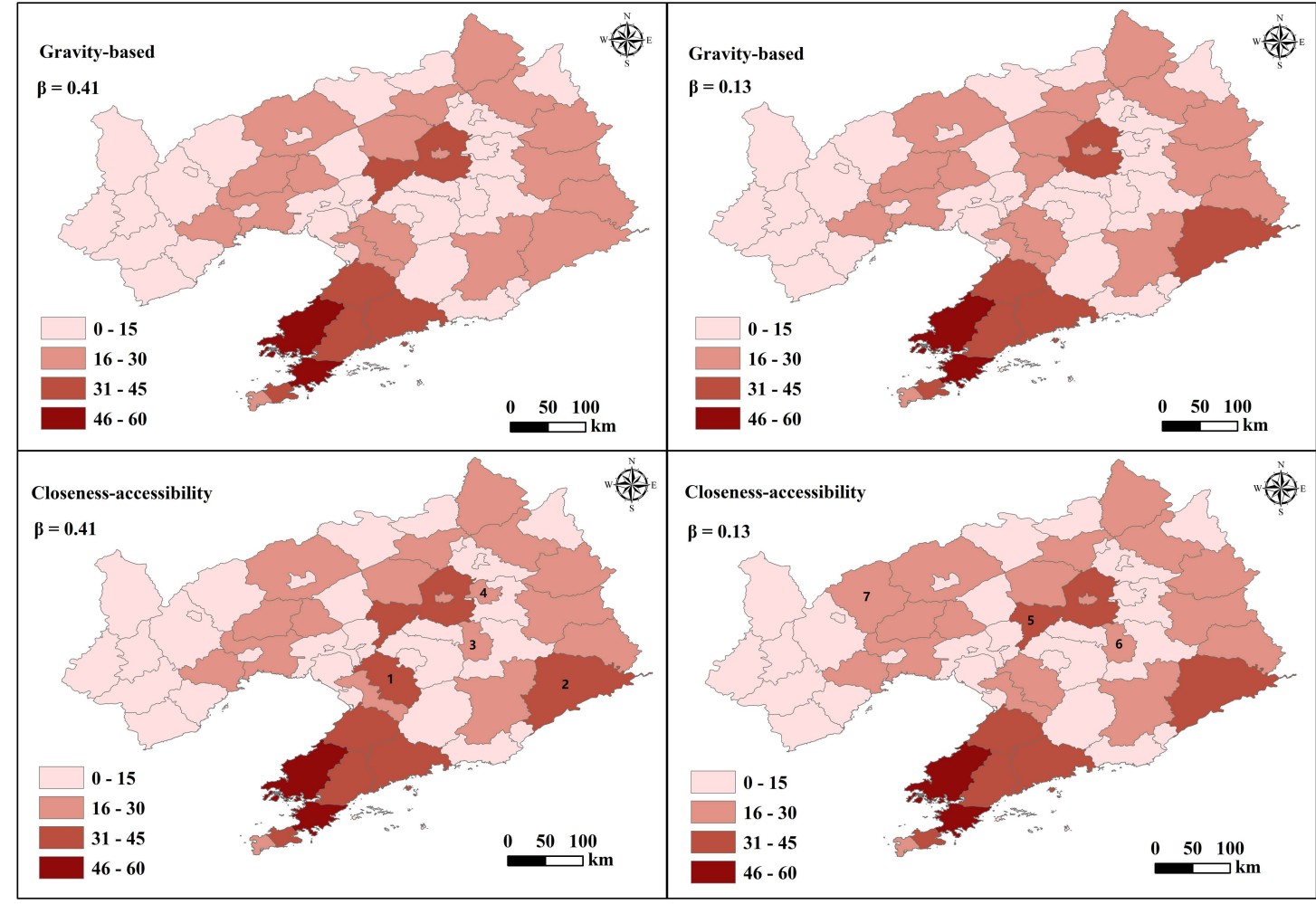

**Fig 7. Gravity-based and closeness-accessibility of regional demand points.** Base map data from GADM (https://gadm.org), version 4.0.

levels and accessibility levels are more prevalent in the eastern region. Discrepancies in car ownership and accessibility levels are concentrated in the western and northeastern regions. The average proportion of regions with low road network accessibility is 21.06%, indicating insufficient accessibility services.

Fig 10 and Fig 11 display the accessibility levels of national, provincial, and county highway networks, comparing the gravity-based and closeness-accessibility with varying β values. The β value's magnitude indicates the extent of traffic impedance's impact on accessibility, with higher values signifying greater influence. Fig 10 shows different β values result in differences in the accessibility of the highway network. Meanwhile, Fig 11 shows the accessibility of various types of highways. Approximately 13% of the province lacks national highways, with 9% lacking provincial highways and 10% lacking county highways, primarily concentrated in the southeast island area and northwest region. National highways with high accessibility are situated in coastal areas and the provincial capital's center, while provincial highways with high accessibility are found in the capital's vicinity. County highways with high accessibility are located in the provincial capital and southwest regions, whereas those with low accessibility are concentrated in western Liaoning. The results are consistent with the results of the highway network accessibility shown in Fig 9.

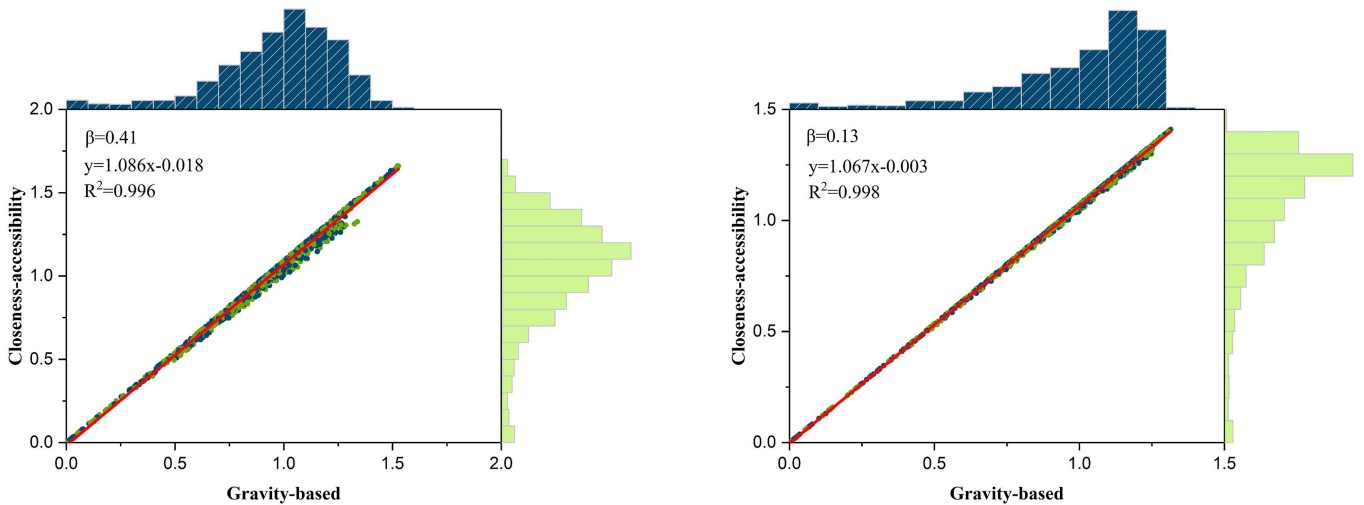

**Fig 8. Gravity-based and closeness-accessibility of the highway network with histograms.**

In this study, when the accessibility level exceeds 0.9, it is classified as high-level accessibility. Table 1 shows the percentage of highways with high-level accessibility at varying β values When β = 0.41, the obtained result is relatively small, indicating that travel cost is an important factor in the accessibility measurement. While when β = 0.13, it shows a small impact on accessibility, suggesting that the β value has an elastic regulatory effect on accessibility.

Closeness-accessibility results surpass those of the gravity-based, indicating that higher trip costs lead to decreased highway network accessibility [73]. The average proportions of high-level accessibility for national, provincial, and county highways are 83.95%, 81.24%, and 82.45%, respectively, underscoring the high service level across all highway grades. However, within the three highway classes, national highways exhibit the lowest accessibility, followed by provincial highways and county highways, with average proportions of 28.41%, 31.57%, and 40.02%, respectively, demonstrating that higher-grade highways sometimes show lower accessibility [74–76].

## Discussion

The impact of trip demand on the accessibility of the highway network: Different types of trip demands(such as education, medical treatment, tourism, etc.) have different functional requirements for the highway network, which is an important basis for highway network planning and design. Trip demands can improve the construction scale, layout and grade structure of the road network, thereby enhancing its accessibility [77,78]. The closeness-accessibility approach proposed in this study considers the number of demand points and the centrality of demand attraction points. It improves the calculation of the shortest path between demand points, enhances the accuracy of accessibility calculations, and provides a clear depiction of accessibility levels across regions. By incorporating trip demand considerations, it is possible to strategically plan the construction and operational costs of the highway network to achieve economic sustainability.

The non-uniform impact of network node centrality on highway network accessibility: Node centrality is commonly utilized to measure the significance or impact of nodes in a network. In highway networks, nodes with high centrality typically exhibit enhanced accessibility due to their closer connections to other nodes, facilitating smoother traffic [69]. This study combines the idea of near analysis, takes into account the influence of centrality on the potential interactions between nodes, introduces the improved gravity-based model with closeness centrality, analyzes the non-uniform influence of the centrality of each pair of nodes, and generates a non-uniform accessibility distribution. Nodes with high centrality are surrounded by areas with heightened accessibility, while nodes with low centrality are surrounded by areas with reduced

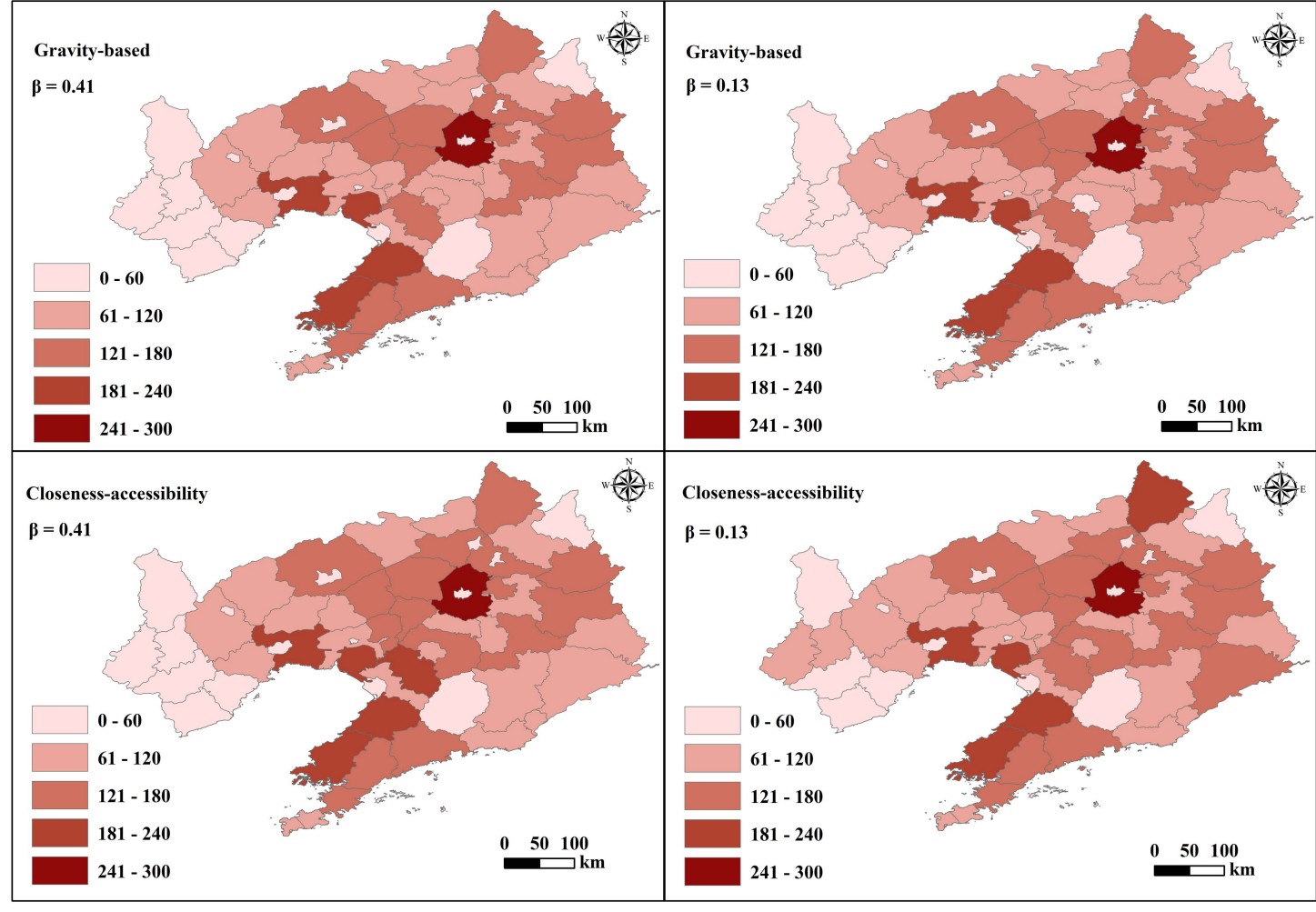

**Fig 9. Gravity-based and closeness-accessibility of regional highway network.** Base map data from GADM (https://gadm.org), version 4.0.

accessibility. The uneven distribution of regional accessibility interacts with and complements economic and social development.

The difference from betweenness-accessibility: Betweenness centrality is the number of paths connecting other nodes that pass through it. The betweenness of a node is a measure of the frequency with which the node is part of the shortest paths that connect all pairs of nodes in the network [53]. It describes the control ability of the node pair in the network over the transmission of information along the shortest path. The higher the value, the stronger the control power of the node pair over information dissemination. Closeness centrality assesses the closeness of a node to others by calculating the reciprocal of the average shortest path distance between the node and all other nodes in the network. The smaller the value, the closer the node is to other nodes.

The two approaches are respectively combined with the gravity-based model. Both introduce a weight to measure the interaction potential, which can overcome the shortcomings of the traditional betweenness metric discovered previously. However, there are differences. the concept of betweenness-accessibility [55] helps to provide a richer picture of the ways a transportation system operates to generate connectivity. It allows a quantification of accessibility to take place in a

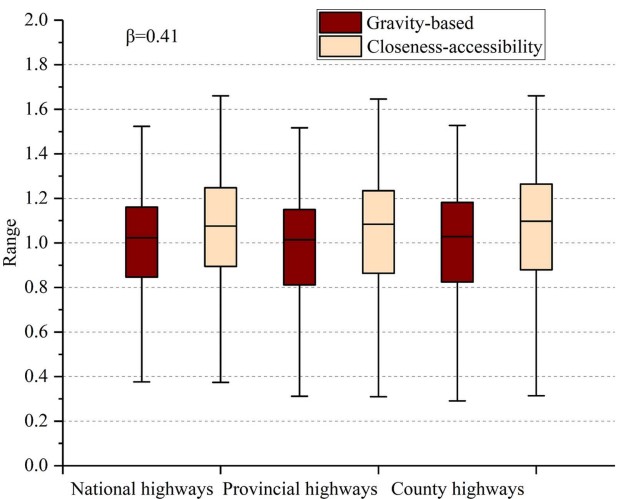
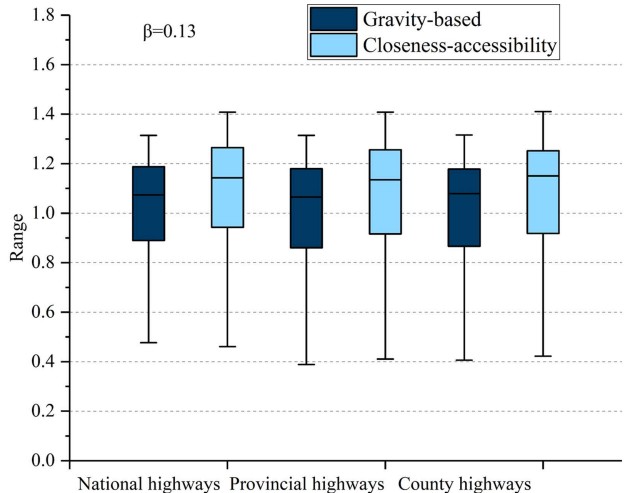

**Fig 10. Accessibility distribution of tertiary highways.**

network-based way; a perspective which can find application inareas such as spatial- and social-equity analysis, vulnerability analysis. Closeness-accessibility is used to analyze the non-uniform impact of a node's centrality, with a focus on studying the accessibility of transportation systems that combine time and space.

Correlation analysis of higher-grade highways and Lower accessibility: The seemingly contradictory phenomenon that higher-grade highways sometimes show lower accessibility is a core paradox in transportation geography and urban planning. Higher-grade highways connect the road networks at the entrances and exits of cities, forming the external transportation channels of cities, and their main transportation function lies in trafficability. Literature [32] defines the concept of accessibility as "the convenience of reaching any land use activity from a specific location using a particular transportation system". Higher accessibility of highways can be interpreted as the higher convenience of reaching other points from a point in the highway transportation system. It is evident that the attributes and geographical locations of origin and destination nodes directly affect the results of accessibility analysis of the highway network. This paper focuses on analyzing the accessibility between demand points (administrative center, comprehensive transportation, education and healthcare, and cultural and tourism). The geographical locations of demand points are mainly concentrated in cities. In contrast, county highways have higher accessibility scores due to their superior transportation functions, while national highways have lower accessibility. This indicates that higher-grade highways sometimes show lower accessibility [74–76].

## Conclusions

In this paper, a new set of indicators combining the concepts of closeness centrality and gravity-based accessibility is introduced. In particular, the indicator expanded the potential of both centrality analysis and accessibility analysis. From the network analysis perspective, closeness-accessibility introduces a weight that measures interaction potential, encompassing a comprehensive spatial-temporal accessibility analysis. In conclusion, the new centrality formulation allows overcoming previously identified shortcomings of traditional accessibility analysis method, addressing some of the limitations of existing models that ignore the influence of network topology and node centrality on spatial accessibility.

The study aims to investigate the impact of travel demand on highway network accessibility and the significance of node centrality non-uniformity. To achieve this, a neighbor demand network is constructed through near analysis. Consider the median and mean of the trip time to calculate different β values, and compare and analyze the gravity-based and closeness-accessibility. Based on the accessibility scale of the highway network at the demand attraction points, determine the accessibility levels of various highways in the study areas.

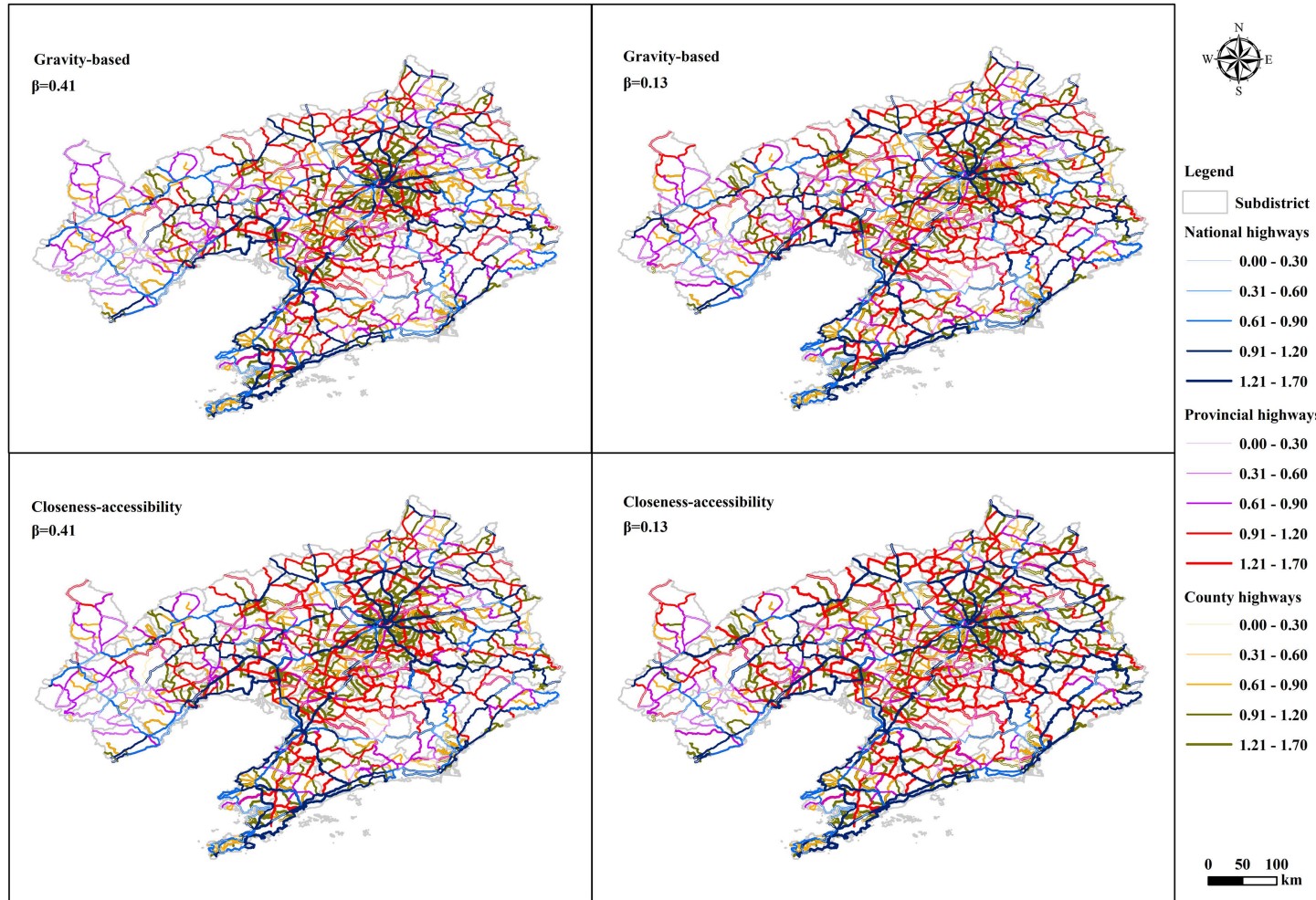

**Fig 11. Accessibility of the highway network at all levels.**

**Table 1. Percentage of high-level accessibility of highways.**

|  | β = 0.41 | | β = 0.13 | |
|---|---|---|---|---|
|  | **Gravity-based** | **Closeness-accessibility** | **Gravity-based** | **Closeness-accessibility** |
| National highways | 79.89 | 83.64 | 84.55 | 87.70 |
| Provincial highways | 75.82 | 80.35 | 82.55 | 86.24 |
| County highways | 78.77 | 81.58 | 83.55 | 85.91 |

The results show that the construction of the highway neighbor demand network in Liaoning Province contains 4,605 nodes and 5,460 connected edges, with the weight being the travel time, and the network is a connected network. There is a positive linear correlation between the gravity-based and closeness-accessibility produced by β = 0.41 and β = 0.13. Specifically, a higher trip cost is associated with reduced accessibility within the highway network. Enhancing the accuracy of accessibility calculations can be achieved through determining the shortest path of the nodes' spatial positions.

Moreover, the centrality distribution of network nodes approximates a normal distribution, indicating a non-uniform impact on highway network accessibility. In Liaoning Province, demand attraction points and areas with high highway network accessibility are predominantly clustered in coastal regions and provincial capital centers, with inadequate accessibility services in western Liaoning. The average proportions of high-grade accessibility for national, provincial, and county highways are 83.95%, 81.24%, and 82.45%, respectively, underscoring the high service level across all highway grades. However, within the three highway classes, national highways exhibit the lowest accessibility, followed by provincial highways and county highways. The higher-grade highways sometimes show lower accessibility.

In future work, we will analyze the estimation method of speed on adjacent arc segments to enhance the speed differentials among various links in the neighbor demand network. This approach aims to obtain a more comprehensive assessment of link trip times, thereby rectifying the results of the current study. Additionally, we will adjust the calculation methodology for the β parameter to facilitate comparisons across different values and analyze the elastic impact of traffic impedance on accessibility. We will also study dynamic and real-time accessibility modeling, which use time-sensitive data or mobile traces to better capture temporal variability in accessibility. Concurrently, a comparative and sensitivity analysis of weighting methods used to evaluate the service capacity of computing facility points will be performed, with the aim of improving the scientific validity and reproducibility of the study. This research offers a methodological tool for transportation experts. It is possible to better guide the construction of various levels of road networks by assessing highway network accessibility.

## Author contributions

**Data curation:** Jinguang Sun, Peng Dai.

**Formal analysis:** Yuanyuan Zhang, Jinguang Sun, Peng Dai.

**Funding acquisition:** Weidong Song.

**Investigation:** Yuanyuan Zhang.

**Methodology:** Yuanyuan Zhang, Weidong Song, Peng Dai.

**Project administration:** Jinguang Sun.

**Supervision:** Jinguang Sun.

**Validation:** Jinguang Sun.

**Visualization:** Weidong Song.

**Writing – original draft:** Yuanyuan Zhang, Jinguang Sun, Peng Dai.

**Writing – review & editing:** Yuanyuan Zhang, Peng Dai.

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
