## [Decision Letter · Decision Letter 0]

10 Jul 2025

Dear Dr. Song,

Thank you for submitting your manuscript to PLOS ONE. After careful consideration, we feel that it has merit but does not fully meet PLOS ONE’s publication criteria as it currently stands. Therefore, we invite you to submit a revised version of the manuscript that addresses the points raised during the review process.

We look forward to receiving your revised manuscript.

Kind regards,

Qing-Chang Lu

Academic Editor

PLOS ONE

Journal Requirements:

3. Please ensure that you include a title page within your main document. We do appreciate that you have a title page document uploaded as a separate file, however, as per our author guidelines (http://journals.plos.org/plosone/s/submission-guidelines#loc-title-page) we do require this to be part of the manuscript file itself and not uploaded separately.

“National Natural Science Foundation of China 42071343”

5. Thank you for uploading your study's underlying data set. Unfortunately, the repository you have noted in your Data Availability statement does not qualify as an acceptable data repository according to PLOS's standards.

6. PLOS requires an ORCID iD for the corresponding author in Editorial Manager on papers submitted after December 6th, 2016. Please ensure that you have an ORCID iD and that it is validated in Editorial Manager. To do this, go to ‘Update my Information’ (in the upper left-hand corner of the main menu), and click on the Fetch/Validate link next to the ORCID field. This will take you to the ORCID site and allow you to create a new iD or authenticate a pre-existing iD in Editorial Manager.

7. Please ensure that you include a title page within your main document. You should list all authors and all affiliations as per our author instructions and clearly indicate the corresponding author.

8. We note that Figures 4,7 and 9 in your submission contain map/satellite images which may be copyrighted. All PLOS content is published under the Creative Commons Attribution License (CC BY 4.0), which means that the manuscript, images, and Supporting Information files will be freely available online, and any third party is permitted to access, download, copy, distribute, and use these materials in any way, even commercially, with proper attribution. For these reasons, we cannot publish previously copyrighted maps or satellite images created using proprietary data, such as Google software (Google Maps, Street View, and Earth). For more information, see our copyright guidelines: http://journals.plos.org/plosone/s/licenses-and-copyright.

 1. You may seek permission from the original copyright holder of Figures 4,7 and 9  to publish the content specifically under the CC BY 4.0 license. 

Additional Editor Comments:

Reviewers' comments:

Reviewer's Responses to Questions

**Comments to the Author**

1. Is the manuscript technically sound, and do the data support the conclusions?

Reviewer #1: Yes

Reviewer #2: Partly

Reviewer #3: Yes

2. Has the statistical analysis been performed appropriately and rigorously?

Reviewer #1: Yes

Reviewer #2: Yes

Reviewer #3: Yes

3. Have the authors made all data underlying the findings in their manuscript fully available?

Reviewer #1: Yes

Reviewer #2: Yes

Reviewer #3: Yes

4. Is the manuscript presented in an intelligible fashion and written in standard English?

Reviewer #1: Yes

Reviewer #2: Yes

Reviewer #3: Yes

Reviewer #1: 1.The paper introduces a novel metric, closeness-accessibility, that extends traditional gravity-based accessibility by integrating closeness centrality from network theory.The authors should more clearly differentiate this metric from similar network-based accessibility metrics found in recent studies (e.g., betweenness-adjusted gravity models).

2.More justification is needed for the specific choice of β values (0.13 and 0.41). Are these empirically derived, or do they stem from prior studies?

3.GIS and Python (NetworkX) are appropriate tools, their implementation details (e.g., algorithms for shortest paths or centrality) need to be described more precisely.

4.Include more detailed maps showing actual road segments and accessibility zones, not just interpolated surfaces.

5.The interpretation of inverse relationships (e.g., lower accessibility with higher road grades) needs further discussion—why do higher-grade roads sometimes show lower accessibility?

6.In the limitations section, discuss the lack of dynamic travel data (e.g., congestion, real-time demand fluctuations) which will significantly influence accessibility.

7.There are some grammatical errors and long sentences it needs to be corrected.

Reviewer #2: The manuscript titled “Accessibility measurement of highway transportation networks based on closeness-accessibility” presents a novel methodological contribution by integrating closeness centrality into a gravity-based model to develop a new accessibility metric. The proposed “closeness-accessibility” approach introduces network science principles into traditional transport geography metrics and is implemented in the context of Liaoning Province, China. This integration is conceptually sound and addresses some of the limitations of existing models that ignore the influence of network topology and node centrality on spatial accessibility.

The paper is strong in its methodological structure. The workflow, including GIS-based near analysis, entropy-based weighting of socio-economic indicators (population, GDP, car ownership), and network construction using Python’s NetworkX library, is rigorous and replicable. The case study is data-rich and effectively demonstrates the empirical value of the proposed metric. The comparison of the closeness-accessibility model with the classical gravity-based model across multiple impedance coefficients (β values) further strengthens the validation process, and the inclusion of visual comparisons through maps and histograms enhances clarity.

However, several areas require improvement. First, the literature review would benefit from a more critical synthesis of previous accessibility models. While the paper reviews the types of accessibility (infrastructure-based, location-based, individual-based, and utility-based), it does not explicitly discuss how closeness-accessibility addresses the specific limitations of earlier gravity-based or centrality-only approaches. Moreover, the review could be strengthened by referencing recent developments in dynamic and real-time accessibility modeling, which use time-sensitive data or mobile traces to better capture temporal variability in accessibility.

Second, the paper does not sufficiently acknowledge the assumptions and limitations of the proposed model. For instance, the calculation of trip time does not account for congestion, road quality, or temporal traffic variability, all of which significantly affect real-world accessibility. Similarly, while entropy weighting is a valid technique, its impact on final accessibility scores is not examined through sensitivity analysis. Clarifying the choice of weighting method and discussing alternatives like AHP or equal weighting would improve transparency and reproducibility.

Third, while the modified equations for closeness centrality in disconnected graphs (Equations 4 and 5) are mentioned, the manuscript does not provide illustrative examples or diagnostics for such subnetworks. Readers would benefit from a brief explanation or visualization showing how centrality scores are normalized across disconnected components, especially since the study uses a large network with heterogeneous connections.

Another concern lies in the interpretation of results. The finding that county highways have higher accessibility scores than provincial or national highways contradicts typical assumptions about infrastructure hierarchy. The authors should further explore whether this result reflects true functional accessibility or is influenced by the spatial concentration of county roads in urbanized regions. Without this clarification, the practical implications of the finding remain ambiguous.

In terms of presentation, terminology is at times inconsistent (e.g., “closeness-accessibility” vs. “closeness centrality-accessibility”), and some figures and captions require more detail. For instance, the distributions in Fig. 5 should clarify whether they represent the entire network or subsets. The manuscript would also benefit from clearer language in some sections to avoid redundancy and enhance readability. It is also recommended to ensure that figures are interpretable in grayscale and accessible to colorblind readers.

In conclusion, the proposed closeness-accessibility framework offers a significant step forward in accessibility modeling by bridging spatial interaction theory and network science. The approach is methodologically sound and holds promise for transport planning applications. However, before publication, the manuscript should address the noted concerns regarding literature framing, model assumptions, and result interpretation. Subject to these minor revisions, the paper has strong potential for publication and could be a valuable resource for researchers and practitioners in transportation geography and urban planning.

Reviewer #3: 1. Add section on the objectives of the research after introduction section clearly stating the need, Objectives, Scope and limitations of the study.

2. Check all the figures with GIS Maps for scale - All figures should be on same scale. Some of the maps are distorted.

3. Conclusions should be supported by data from results and analysis section, revise the conclusion section accordingly

**Do you want your identity to be public for this peer review?** For information about this choice, including consent withdrawal, please see our Privacy Policy

Reviewer #1: No

Reviewer #2: No

Reviewer #3: **Yes: ** Tejwant Singh Brar

---

## [Author Response · Author response to Decision Letter 1]

23 Aug 2025

We thank all the editors and reviewers for their valuable comments and suggestions. We have carefully revised the manuscript to enhance its clarity and facilitate the understanding of the readers. Our point-to-point responses are presented in the following. We hope that the revision would satisfactorily address the comments and concern of the editors and reviewers.

Editor comments

Thank you very much for handing our manuscript(ID: PONE-D-25-21841)! We have revised the paper according to your and Reviewers’ comments. The summary of the major revisions we made is as follows:

1. Thanks you for the comment regarding the style requirements of the manuscript. We have added a title page and made some adjustments to the manuscript according to the PLOS ONE style templates.

2. Thank you for your feedback regarding the data set. We have now uploaded the minimal dataset underlying our study to the figshare repository, which complies with PLOS data availability standards. The dataset can be accessed via the following DOI: 10.6084/m9.figshare.29973184.

3. I have an ORCID iD and that it is validated in Editorial Manager.

4. Thank you for the comment regarding the copyrighted maps. We have made some changes to the figures 4,7,9 of the manuscript according to Reviewer #3’ comments with GIS Maps.

5.This amended Role of Funder statement: "The funders had no role in study design, data collection and analysis, decision to publish, or preparation of the manuscript."

6. We have fine-tuned and optimized the English expression throughout the text.

Reviewer #1:

1.The paper introduces a novel metric, closeness-accessibility, that extends traditional gravity-based accessibility by integrating closeness centrality from network theory.The authors should more clearly differentiate this metric from similar network-based accessibility metrics found in recent studies (e.g., betweenness-adjusted gravity models).

Response Thanks for the comment. As your suggestion, adding the difference closeness-accessibility and betweenness-accessibility. We have re-optimized the accessibility analysis methods and discussion section. The adjustment of accessibility analysis methods section is in line 162-175 page 8, and we also copy here:

“……Centrality analysis is a key issue in network studies and serves as a method to assess the significance of nodes within a network. The commonly used centrality measure is the degree centrality of a node, which quantifies the number of edges connected to a node [52]. Betweenness centrality evaluates the number of shortest paths that pass through a node [53]. Closeness centrality measures how close a node is to other nodes [54]. In network analysis, centrality is related to the paths of node interactions, and various centrality perspectives can supplement the accessibility analysis [55-57].……”

“……Georgios et al. (2020) [55] propose a new centrality measure called betweenness-accessibility, which is useful to estimate the impacts of accessibility on networks as potential for interaction is reflected on them. The new centrality formulation allows overcoming previously identified shortcomings of traditional betweenness measures, resulting in a measure tailored for networks with heterogeneous interaction levels. A new set of indicators combining the concepts of centrality and gravity-based accessibility was introduced and expanded the potential of both centrality analysis and accessibility analysis.……”

The adjustment of discussion section is in line 510-526 page 29-30, and we also copy here:

“……The difference from betweenness-accessibility: Betweenness centrality is the number of paths connecting other nodes that pass through it. the betweenness of a node is a measure of the frequency with which the node is part of the shortest paths that connect all pairs of nodes in the network [53]. It describes the control ability of the node pair in the network over the transmission of information along the shortest path. The higher the value, the stronger the control power of the node pair over information dissemination. Closeness centrality assesses the closeness of a node to others by calculating the reciprocal of the average shortest path distance between the node and all other nodes in the network. The smaller the value, the closer the node is to other nodes.……”

“……The two approaches are respectively combined with the gravity-based model. Both introduce a weight to measure the interaction potential, which can overcome the shortcomings of the traditional betweenness metric discovered previously. However, there are differences. the concept of betweenness-accessibility [55] helps to provide a richer picture of the ways a transportation system operates to generate connectivity. It allows a quantification of accessibility to take place in a network-based way; a perspective which can find application inareas such as spatial- and social-equity analysis, vulnerability analysis. Closeness-accessibility is used to analyze the non-uniform impact of a node's centrality, with a focus on studying the accessibility of transportation systems that combine time and space.……”

2.More justification is needed for the specific choice of β values (0.13 and 0.41). Are these empirically derived, or do they stem from prior studies?

Response Thanks for the comment. As your suggestion, β values are empirically derived, or stem from prior studies. Due to the reliance on subjectivity in empirical inference, we adopt a simple calculation method to obtain parameter values. We have rewritten and modified the section in line 237-241 page 13, and in line 306-308 page 16:

“……The calibration of the β parameter is a challenge when computing gravity models, The literature recommends an iterative process based on observed trip cost data to calibrate this parameter for gravity models [58]. However, such data is currently unavailable, making the calibration process unfeasible. The method used by Östh et al. (2014) [59] for calculating the β value is cited, as shown in Eq.3.……”

“……The calculation method of the β value is as shown in Eq. 3. This study calculates various β values for the median and mean of trip time to obtain comparable accessibility analysis results under the influence of demand elasticity..……”

3.GIS and Python (NetworkX) are appropriate tools, their implementation details (e.g., algorithms for shortest paths or centrality) need to be described more precisely.

Response Thanks for the comment. As your suggestion, We have added the implementation details of algorithms for shortest paths and centrality in line 368-370 page 20, and we also copy here:

“……The Dijkstra algorithm is used to obtain the shortest path distances between all nodes in the network, and Eq.4 is used to calculate the closeness centrality of the nodes in the highway neighbor demand network.……”

4.Include more detailed maps showing actual road segments and accessibility zones, not just interpolated surfaces.

Response Thanks for the comment. We appreciate your suggestion regarding more detailed maps with actual road segments and accessibility zones. The current maps are generated using interpolated surfaces, which provide a generalized view of accessibility, as depicted in Fig 9. While we agree that including actual road segments and accessibility zones would enhance the maps, the current map displays the accessibility levels of national, provincial, and county highway networks, as depicted in Fig 11. We hope this still meets your needs.

5.The interpretation of inverse relationships (e.g., lower accessibility with higher road grades) needs further discussion—why do higher-grade roads sometimes show lower accessibility?

Response Thanks for the comment. As your suggestion, we have re-optimized the discussion section to interpret the correlation of higher-grade roads and lower accessibility in line 527-542 page 30-31, and we also copy here:

“……Correlation analysis of higher-grade highways and Lower accessibility: The seemingly contradictory phenomenon that higher-grade highways sometimes show lower accessibility is a core paradox in transportation geography and urban planning. Higher-grade highways connect the road networks at the entrances and exits of cities, forming the external transportation channels of cities, and their main transportation function lies in trafficability. Literature [32] defines the concept of accessibility as "the convenience of reaching any land use activity from a specific location using a particular transportation system". Higher accessibility of highways can be interpreted as the higher convenience of reaching other points from a point in the highway transportation system. It is evident that the attributes and geographical locations of origin and destination nodes directly affect the results of accessibility analysis of the highway network. This paper focuses on analyzing the accessibility between demand points (administrative center, comprehensive transportation, education and healthcare, and cultural and tourism). The geographical locations of demand points are mainly concentrated in cities. In contrast, county highways have higher accessibility scores due to their superior transportation functions, while national highways have lower accessibility. This indicates that higher-grade highways sometimes show lower accessibility [74-76].……”

6.In the limitations section, discuss the lack of dynamic travel data (e.g., congestion, real-time demand fluctuations) which will significantly influence accessibility.

Response Thanks for the comment. As your suggestion, we have added the assumptions and limitations of the model. The research methodology does not involve dynamic and real-time accessibility modeling, that will serve as the research focus for future work. The assumptions and limitations of the model are in line 283-285 page 15, and we also copy here:

“……The assumptions and limitations of the model are that the calculation of trip time does not take into account for the influence of dynamic and real-time data such as congestion, road quality, temporal traffic variability, driving habits, weather, etc.…….”

The future work is in line 578-580 page 32, and we also copy here:

“……We will also study dynamic and real-time accessibility modeling, which use time-sensitive data or mobile traces to better capture temporal variability in accessibility.…….”

7.There are some grammatical errors and long sentences it needs to be corrected.

Response Thanks for the comment. As your suggestion, we have fine-tuned and optimized the English expression throughout the text, and we apologize for the disturbance caused to your reading by the long sentences.

Reviewer #2:

The manuscript titled “Accessibility measurement of highway transportation networks based on closeness-accessibility” presents a novel methodological contribution by integrating closeness centrality into a gravity-based model to develop a new accessibility metric. The proposed “closeness-accessibility” approach introduces network science principles into traditional transport geography metrics and is implemented in the context of Liaoning Province, China. This integration is conceptually sound and addresses some of the limitations of existing models that ignore the influence of network topology and node centrality on spatial accessibility.

The paper is strong in its methodological structure. The workflow, including GIS-based near analysis, entropy-based weighting of socio-economic indicators (population, GDP, car ownership), and network construction using Python’s NetworkX library, is rigorous and replicable. The case study is data-rich and effectively demonstrates the empirical value of the proposed metric. The comparison of the closeness-accessibility model with the classical gravity-based model across multiple impedance coefficients (β values) further strengthens the validation process, and the inclusion of visual comparisons through maps and histograms enhances clarity.

Response We sincerely thank you for the valuable feedback that we have used to improve the quality of our manuscript.

1.the literature review would benefit from a more critical synthesis of previous accessibility models. While the paper reviews the types of accessibility (infrastructure-based, location-based, individual-based, and utility-based), it does not explicitly discuss how closeness-accessibility addresses the specific limitations of earlier gravity-based or centrality-only approaches. Moreover, the review could be strengthened by referencing recent developments in dynamic and real-time accessibility modeling, which use time-sensitive data or mobile traces to better capture temporal variability in accessibility.

Response Thanks for the comment. As you mentioned, We have combed the literature review section of the article. The research methodology does not involve dynamic and real-time accessibility modeling, that will serve as the research focus for future work. We have added the section on the objectives of the research after introduction section clearly stating the need, ibjectives, scope and limitations of the study. Thank you again for your suggestions. Hence, we had added a passage of the introduction section in line 80-85 page 4 , and we also copy here:

“……In this paper, we are interested in the potential impacts of accessibility on networks. The proposed closeness-accessibility modeling bridge spatial interaction theory and network science. The study aims to investigate the impact of travel demand on highway network accessibility and the significance of node centrality non-uniformity. Our case study focused on the three-level highway network of national, provincial and county highways in Liaoning province, utilizing only structural data and excluding any information about traffic flow.……”

we had added a passage of the literature review section in line 162-175 page 8 , and we also copy here:

“……Centrality analysis is a fundamental aspect of network studies, serving as a method to assess the significance of nodes within a network. Degree centrality, a commonly employed metric, quantifies the number of edges connected to a node [52]. Betweenness centrality, on the other hand, evaluates the number of shortest paths that pass through a node [53]. Closeness centrality measures how close a node is to other nodes [54]. The concept of centrality in network analysis is closely tied to the flow of interactions among nodes, and various centrality perspectives can enhance the understanding of network accessibility [55-57].……”

“……Georgios et al. (2020) [55]propose a new centrality measure called betweenness-accessibility, which is useful to estimate the impacts of accessibility on networks as potential for interaction is reflected on them. The new centrality formulation allows overcoming previously identified shortcomings of traditional betweenness measures, resulting in a measure tailored for networks with heterogeneous interaction levels. A new set of indicators combining the concepts of centrality and gravity-based accessibility was introduced and expanded the potential of both centrality analysis and accessibility analysis.……”

The future work in line 578-580 page32 , and we also copy here:

“……We will also study dynamic and real-time accessibility modeling, which use time-sensitive data or mobile traces to better capture temporal variability in accessibility.……”

2.the paper does not sufficiently acknowledge the assumptions and limitations of the proposed model. For instance, the calculation of trip time does not account for congestion, road quality, or temporal traffic variability, all of which significantly affect real-world accessibility. Similarly, while entropy weighting is a valid technique, its impact on final accessibility scores is not examined through sensitivity analysis. Clarifying the choice of weighting method and discussing alternatives like AHP or equal weighting would improve transparency and reproducibility.

Response Thanks for the comment. As you suggest, adding the assumptions and limitations of the model here is necessary. Therefore, we have added the assumptions and limitations of the model and copied them below. We agree that the selection of the entropy weighting should be clarified, and sensitivity analysis can help improve the robustness of

---

## [Decision Letter · Decision Letter 1]

1 Oct 2025

Dear Dr. Song,

Thank you for submitting your manuscript to PLOS ONE. After careful consideration, we feel that it has merit but does not fully meet PLOS ONE’s publication criteria as it currently stands. Therefore, we invite you to submit a revised version of the manuscript that addresses the points raised during the review process.

We look forward to receiving your revised manuscript.

Kind regards,

Qing-Chang Lu

Academic Editor

PLOS ONE

**Journal Requirements:**

**Additional Editor Comments:**

Please address the minor comments from the reviewer. Thanks.

Reviewers' comments:

Reviewer's Responses to Questions

**Comments to the Author**

Reviewer #1: All comments have been addressed

Reviewer #2: All comments have been addressed

Reviewer #3: (No Response)

2. Is the manuscript technically sound, and do the data support the conclusions?

Reviewer #1: Yes

Reviewer #2: Yes

Reviewer #3: Yes

3. Has the statistical analysis been performed appropriately and rigorously?

Reviewer #1: Yes

Reviewer #2: Yes

Reviewer #3: Yes

4. Have the authors made all data underlying the findings in their manuscript fully available?

Reviewer #1: Yes

Reviewer #2: Yes

Reviewer #3: Yes

5. Is the manuscript presented in an intelligible fashion and written in standard English?

Reviewer #1: Yes

Reviewer #2: Yes

Reviewer #3: Yes

**Reviewer #1:**  (No Response)

**Reviewer #2: ** It seems authors addressed previous comments. I would like to recommend to published articles after minor improvements, such as improving resolution of imagers

**Reviewer #3:**  1. Still the section on the objectives of the research after introduction section clearly stating the need, Objectives, Scope and limitations of the study. In line 80-85 Page 17 in introduction section what you have written is Aim of research which is general which objectives are specific.

**Do you want your identity to be public for this peer review?** For information about this choice, including consent withdrawal, please see our Privacy Policy

Reviewer #1: No

Reviewer #2: **Yes: ** Amila Jayasinghe

Reviewer #3: **Yes: ** Tejwant Singh Brar

---

## [Author Response · Author response to Decision Letter 2]

24 Oct 2025

Submission ID: PONE-D-25-21841

Title: Accessibility measurement of highway transportation networks based on closeness-accessibility

We thank all the editors and reviewers for their valuable comments and suggestions. We have carefully revised the manuscript to enhance its clarity and facilitate the understanding of the readers. Our point-to-point responses are presented in the following. We hope that the revision would satisfactorily address the comments and concern of the editors and reviewers.

Reviewer #1:

We sincerely thank you for the valuable feedback that we have used to improve the quality of our manuscript.

Reviewer #2:

It seems authors addressed previous comments. I would like to recommend to published articles after minor improvements, such as improving resolution of imagers.

Response We are very grateful to the reviewer for the positive evaluation and the recommendation for publication. We also appreciate the constructive feedback on the image quality. Following the reviewer’s suggestion, we have carefully improved the resolution of all figures in the manuscript to ensure they are clear and meet the publication standards. All figures have been regenerated and exported at a high resolution (600 DPI) by the Preflight Analysis and Conversion Engine (PACE) digital diagnostic tool.

We have also taken this opportunity to proofread the entire manuscript again to correct any minor typographical errors. Thank you once again for your valuable time and guidance.

Reviewer #3:

We sincerely thank you for the valuable feedback that we have used to improve the quality of our manuscript.The comments are laid out below in bold font and specific concerns have been numbered. Our response is given in purple font and changes to the manuscript are highlighted in yellow.

1. Still the section on the objectives of the research after introduction section clearly stating the need, Objectives, Scope and limitations of the study. In line 80-85 Page 17 in introduction section what you have written is Aim of research which is generalwhich objectives are specific.

Response Thanks for the comment. As you suggested, we have adjusted the introduction section of the article to clarify the aim of research in line 80-85 page 4 , and we also copy here:

“……The study aims to quantify the impact of different travel demand patterns on the overall accessibility of the highway network. By comparing network performance in a real network, the importance of node centrality non-uniformity is evaluated. The proposed closeness-accessibility modeling bridge spatial interaction theory and network science. Our case study focuses on the three-level highway network of national, provincial and county highways in Liaoning province, utilizing only structural data and excluding any information about traffic flow.……”

---

## [Editor Report · Decision Letter 2]

3 Nov 2025

Accessibility measurement of highway transportation networks based on closeness-accessibility

PONE-D-25-21841R2

Dear Dr. Song,

We’re pleased to inform you that your manuscript has been judged scientifically suitable for publication and will be formally accepted for publication once it meets all outstanding technical requirements.

Kind regards,

Qing-Chang Lu

Academic Editor

PLOS ONE
---

## [Editor Report · Acceptance letter]

PONE-D-25-21841R2

PLOS ONE

Dear Dr. Song,

I'm pleased to inform you that your manuscript has been deemed suitable for publication in PLOS ONE. Congratulations! Your manuscript is now being handed over to our production team.

Kind regards,

on behalf of

Dr. Qing-Chang Lu

Academic Editor

PLOS ONE